# Seasonal Variation in the Rhizosphere and Non-Rhizosphere Microbial Community Structures and Functions of *Camellia yuhsienensis* Hu

**DOI:** 10.3390/microorganisms8091385

**Published:** 2020-09-10

**Authors:** Jun Li, Ziqiong Luo, Chenhui Zhang, Xinjing Qu, Ming Chen, Ting Song, Jun Yuan

**Affiliations:** Key Laboratory of Cultivation and Protection for Non-Wood Forest Trees, Ministry of Education, Central South University of Forestry and Technology, Changsha 410004, China; chuannonglj@163.com (J.L.); luoluoziqiong@163.com (Z.L.); kwinnerkin@163.com (C.Z.); xinjingqu@126.com (X.Q.); hncscm2010@163.com (M.C.); stxiaomaoji@163.com (T.S.)

**Keywords:** *Camellia yuhsienensis* Hu, high-throughput sequencing, seasonal variation, rhizosphere, microorganism

## Abstract

*Camellia yuhsienensis* Hu, endemic to China, is a predominant oilseed crop, due to its high yield and pathogen resistance. Past studies have focused on the aboveground parts of *C. yuhsienensis,* whereas the microbial community of the rhizosphere has not been reported yet. This study is the first time to explore the influence of seasonal variation on the microbial community in the rhizosphere of *C. yuhsienensis* using high-throughput sequencing. The results showed that the dominant bacteria in the rhizosphere of *C. yuhsienensis* were Chloroflexi, Proteobacteria, Acidobacteria, Actinobacteria, and Planctomycetes, and the dominant fungi were Ascomycota, Basidiomycota, and Mucoromycota. Seasonal variation has significant effects on the abundance of the bacterial and fungal groups in the rhizosphere. A significant increase in bacterial abundance and diversity in the rhizosphere reflected the root activity of *C. yuhsienensis* in winter. Over the entire year, there were weak correlations between microorganisms and soil physiochemical properties in the rhizosphere. In this study, we found that the bacterial biomarkers in the rhizosphere were chemoorganotrophic Gram-negative bacteria that grow under aerobic conditions, and fungal biomarkers, such as *Trichoderma*, *Mortierella,* and *Lecanicillium*, exhibited protection against pathogens in the rhizosphere. In the rhizosphere of *C. yuhsienensis*, the dominant functions of the bacteria included nitrogen metabolism, oxidative phosphorylation, glycine, serine and threonine metabolism, glutathione metabolism, and sulfur metabolism. The dominant fungal functional groups were endophytes and ectomycorrhizal fungi of a symbiotroph trophic type. In conclusion, seasonal variation had a remarkable influence on the microbial communities and functions, which were also significantly different in the rhizosphere and non-rhizosphere of *C. yuhsienensis*. The rhizosphere of *C. yuhsienensis* provides suitable conditions with good air permeability that allows beneficial bacteria and fungi to dominate the soil microbial community, which can improve the growth and pathogen resistance of *C. yuhsienensis.*

## 1. Introduction

Microorganisms play a significant role in the decomposition process of soil organic matter, litter, and wood residues [1,2,3]. The microbial community structures partly reflect the nutrient and health conditions of the soil. Some bacterial phyla only proliferate in copiotrophic substances, such as Alphaproteobacteria, Betaproteobacteria, and Bacteroidetes, while others proliferate in oligotrophic substances, such as Acidobacteria [4,5,6]. Some fungi are interdependent with the plant rhizosphere, such as arbuscular mycorrhizal fungi, while others can cause plant disease, such as *Venturia inaequalis* [7,8,9]. Due to their sensitive features, microorganisms are significantly regulated by many factors, such as plant roots and seasonal variation [10,11,12].

The rhizosphere is recognized as one of the most complex environments colonized by significant numbers of microorganisms [13]. Generally, plant roots impact microorganisms via their physiological activity, such as the production of secondary metabolites [14]. Thus, the microbial communities around the rhizosphere have a close association with plant roots and differ significantly from the non-rhizosphere microbial communities. Rhizosphere microorganisms are mainly divided into two categories according to whether they have deleterious or beneficial effects on plants [15]. The beneficial microorganisms, for example, can promote the abiotic stress tolerance of plants, facilitate nutrient absorption and plant growth, and protect plants from pathogens [16,17]. On the contrary, harmful microorganisms, such as deleterious rhizobacteria, are largely saprophytic bacteria that aggressively colonize plant roots, which leads to plant disease, the inhibition of root growth, the suppression of plant growth, and crop failure [18]. Seasonal variation has major direct and indirect influences on the rhizosphere microbial community. Soil water content affects the rhizosphere microbial community significantly. Zhou et al. [19] found that regardless of water availability, microbial activities were restricted in the rhizosphere of wheat. Similarly, the research of Ilyas and Bano [20] indicated that there were more *Azospirillum* strains isolated from the rhizosphere soil of plants in well-watered conditions than in arid and semiarid conditions. Seasonal drought and rainfall have a significant influence on rhizosphere microbial diversity and abundance [21]. Drought stress can indirectly impact microorganisms by restricting plant root growth and regulating root exudation [22]. López-Gutiérrez et al. [23] reported that seasonality could mediate arbuscular mycorrhiza-colonized root length, dehydrogenase activity, and bacterial plate counts in the rhizosphere. Microorganisms are usually classified according to different functions. Due to the activity of plant roots, there are major differences in the microbial functions between the rhizosphere and in the bulk soil. Yang et al. [24] found that microbial catabolic diversity in the rhizosphere differed significantly from that in the bulk soil. Mestre et al. [25] explored yeast community and function in *Nothofagus pumilio* forests. They found that glucose fermentation, organic acid production, and cellobiose assimilation occurred more frequently in the rhizosphere than in the bulk soil. Hernesmaa et al. [26] revealed that the numbers of cultivable bacteria and phosphatase activity in the rhizosphere of Scots pine were significantly higher than in the bulk soil. In summary, microbial communities in the rhizosphere are indispensable for plants [27]. Therefore, researching the dynamics of the rhizosphere microbial community across seasons could inform the promotion of plant growth and yield.

*Camellia yuhsienensis* Hu, a species of oil tea, was once widely cultivated in central China because of its high yield and high resistance to disease pathogens [28]. Cook et al. [29] found that plants tend to regulate microbial communities for their own benefit. Previous studies have reported on the significant carbon metabolic activity in the rhizosphere and bulk soil of *Camellia oleifera* plantations, another species of oil tea [10,30]. Hartmann et al. [31] revealed that specific microenvironments around the rhizosphere lead to specific interactions among microbes with the development of plant roots, resulting in the biological protection of plants from pathogenic microbes. However, the above studies on microbial communities in the rhizosphere or bulk soil of *C. oleifera* plantations did not observe any microorganisms with great disease resistance against pathogens. Previous studies have confirmed that the disease resistance of *C. oleifera* is inferior to that of *C. yuhsienensis* [28,32]. No studies have explored the rhizosphere microbial community of *C. yuhsienensis.* Therefore, we hypothesize that some specific microbial communities must exist that improve the resistance of *C. yuhsienensis* to pathogens, and furthermore, that these microbial communities might be regulated by seasonality. In order to verify this, high-throughput sequencing was used to determine the microbial community structure and functions in the rhizosphere and non-rhizosphere of *C. yuhsienensis* in spring, summer, autumn, and winter.

## 2. Materials and Methods 

### 2.1. Site Description

The sampling site was located at a 10-year-old *C. yuhsienensis* forest in Youxian, Hunan, China (N 27°02′, E 113°20′). The plant and row spacing were 2 m and 3 m, respectively. The climate is subtropical monsoon with a mean annual total rainfall and temperature of 1410 mm and 17.8 °C, respectively [33]. The soil at the experimental site is a Quaternary red clay with a pH of 4.1–4.5. The total organic matter content is 17–23 g/kg. The total nitrogen content is approximately 330–810 mg/kg. The available phosphorus content is 0.1–10 mg/kg, and the available potassium content is 64–95 mg/kg. The chemical properties mentioned above are displayed in Appendix A.

### 2.2. Experimental Design and Sampling

According to the phenological phase of *C. yuhsienensis*, the soil samples were collected on 23 October 2018 (autumn, C), 19 January 2019 (winter, D), 5 April 2019 (spring, A), and 22 July 2019 (summer, B). Three quadrats of 20 m × 20 m were selected in the *C. yuhsienensis* plantation. Five trees were chosen according to an “S” type route in the center of each quadrat. The rhizosphere soil samples of *C. yuhsienensis* were collected, as described by Reference [34] after shoveling the roots from four ordinations approximately 0.5 m away from the trunk. The corresponding non-rhizosphere soil samples at 0–20 cm depth were collected as a control approximately 1 m away from the trunk. The rhizosphere and non-rhizosphere soils in each quadrate were each well blended as a single sample. The soil samples were then transported back to the laboratory on dry ice. After removing debris and roots, the soil samples were well mixed, ground, and sieved (<2 mm). One part of the soil samples was used for evaluating the physiochemical properties after air-drying in the shade, while the remainder was stored at −80 °C for high-throughput sequencing at Genedenovo Biotechnology Co., LTD (Guangzhou, China). The soil temperature at 0–20 cm depth was detected using a Wdsen Electronic temperature recorder (iButton DS1925) during the whole experimental period (Appendix A).

### 2.3. The Measure of Soil Physiochemical Properties

Soil moisture content (SMC) was measured by evaporating the water with burning alcohol [35]. Soil pH, total organic carbon (TOC), total nitrogen (TN), alkaline hydrolyzable nitrogen (AHN), total phosphorous (TP), and total potassium (TK) were measured according to Chinese forestry standards [36,37,38,39,40,41]. A 2 mol/L KCl soil extract was used to measure ammonium nitrogen (AMN) and nitrate nitrogen (NN) using a continuous flow auto-analyzer (Smartchem 200, Westco Scientific Instruments, Roman, Italy). Mehlich 3 extractant was used to extract soil available phosphorous (AP) and potassium (AK), and then AP was measured using a continuous flow auto-analyzer, while AK was measured as TK [42].

### 2.4. Bacterial and Fungal Community Assessment

#### 2.4.1. DNA Extraction and PCR Amplification

Microbial DNA was extracted using HiPure Soil DNA Kits (Magen, Guangzhou, China) according to the manufacturer’s protocols. The 16S rDNA V3-V4 region of the ribosomal RNA gene was amplified by PCR using primers 341F: 5′-CCTACGGGNGGCWGCAG-3′, 806R: 5′-GGACTACHVGGGTATCTAAT-3′. The PCR amplification of 16S r DNA proceeded, as described in Reference [43]. The ITS rDNA region of the ribosomal RNA gene was amplified by PCR using primers ITS3-KYO2 (F): 5′-GATGAAGAACGYAGYRAA-3′ and ITS4 (R): 5′-TCCTCCGCTTATTGATATGC-3′ [44]. The ITS region of the Eukaryotic ribosomal RNA gene was amplified by PCR (95 °C for 2 min, followed by 27 cycles at 98 °C for 10 s, 62 °C for 30 s, and 68 °C for 30 s and a final extension at 68 °C for 10 min). The PCR reactions were performed in triplicate 50 μL mixtures containing 5 μL of 10× KOD Buffer, 5 μL of 2 mM dNTPs, 3 μL of 25 mM MgSO4,1.5 μL of each primer (10 μM), 1 μL of KOD Polymerase, and 100 ng of template DNA.

#### 2.4.2. Illumina Novaseq6000 Sequencing

Amplicons were extracted from 2% agarose gels and purified using the AxyPrep DNA Gel Extraction Kit (Axygen Biosciences, Union City, CA, USA.) according to the manufacturer’s instructions and quantified using an ABI StepOnePlus Real-Time PCR System (Life Technologies, Foster City, CA, USA). Purified amplicons were pooled in equimolar ratios and paired-end sequenced (PE250) on an Illumina platform according to the standard protocols. The raw reads were deposited into the NCBI Sequence Read Archive (SRA) database (BioProject: PRJNA646017).

#### 2.4.3. Statistical and Bioinformatics Analysis

Raw reads were further filtered using FASTP according to the rule: Remove reads containing more than 10% unknown nucleotides (N) and reads with less than 50% of bases with quality (Q-value)>20 [45]. Paired-end clean reads were merged as raw tags using FLASH (version 1.2.11) with a minimum overlap of 10 bp and mismatch error rates of 2% [46]. The effective tags were clustered into operational taxonomic units (OTUs) of ≥ 97% similarity using the UPARSE (version 9.2.64) pipeline [47]. The tag sequence with the highest abundance was selected as a representative sequence within each cluster.

A Venn analysis was used to identify unique and common OTUs among the different groups, was performed using the R “VennDiagram” package (version 1.6.16) [48]. The OTU rarefaction curves and rank abundance curves were plotted using the R “ggplot2” package (version 2.2.1) [49]. Alpha diversities of bacteria and fungi (Sobs, Shannon, and Chao 1 index) were calculated in QIIME (version 1.9.1) [50]. The abundance statistics of each taxonomy were visualized using Krona (version 2.6) [51]. Principal co-ordinates analysis (PCoA) and permutational multivariate analysis of variance (MANOVA) (Permanova) based on unweighted unifrac distance were used to evaluate the influence of plant root and seasonal dynamics on bacterial and fungal community structures. A least discriminant analysis (LDA) effect size (LEfSe) taxonomic cladogram was used to identify bacterial biomarkers (LDA > 3.5) and fungal biomarkers (LDA > 3.5) in the different treatment using LEfSe software [52]. Indicator analysis was used to determine the significantly different species (biomarkers) between rhizosphere and non-rhizosphere soil during the entire year using the R “labdsv” package (version 2.0-1) [53]. The Kyoto Encyclopedia of Genes and Genomes (KEGG) pathway analysis of the OTUs was inferred using Tax4Fun (version 1.0) [54]. The functional group (Guild) of the fungi (relative abundance greater than 0.01% in the rhizosphere) was inferred using FUNGuild (version 1.0) [55]. The bacterial function differences between the rhizosphere and non-rhizosphere samples were calculated by Welch’s *t*-test, and the fungal function differences between the rhizosphere and non-rhizosphere were calculated using the Wilcoxon rank test in the R “vegan” package (version 2.5.3) [56].

One-way analysis of variance (ANOVA) was used to determine the significant differences among different treatments during the whole year. Nonparametric tests with a significance level of 0.05 were used to compare the means of bacterial and fungal alpha diversities in different treatments and seasons. Variance partitioning analysis, generated in the R “vegan” package (version 2.5.3) [56], was used to evaluate the contribution of environmental factors to microbial community structure. Pearson’s correlation coefficients between environmental factors and microbial species and alpha diversities were calculated by Omicshare tools, a free online platform for data analysis [57].

## 3. Results

### 3.1. Soil Microbial Community Structure

#### 3.1.1. Quality Control Report of the High-Throughput Sequencing

The tags and OTUs of the bacteria and fungi are presented in Appendix A and Appendix A. In order to ensure the reproducibility and validity of the microbial data, the data extraction flat and dilution cures were processed before analysis [58]. According to Appendix A, the dilution cure of the Sobs index indicated that there would be more bacteria and fungi if sequencing were continued, but the plateau of the dilution cure of Shannon’s index was reached early, showing that the number of reads was sufficient for this research.

#### 3.1.2. Microbial Composition

According to Figure 1, the dominant bacteria were Chloroflexi, Proteobacteria, Acidobacteria, Actinobacteria, Planctomycetes, Firmicutes, and Bacteroidetes at the phylum level. The dominant fungi were Ascomycota, Basidiomycota, and Mucoromycota at the phylum level. Venn diagrams (Figure 2) indicated that, with the exception of summer, the number of specific bacterial OTUs in the rhizosphere was higher than in the non-rhizosphere. In winter, the number of specific bacterial OTUs in the non-rhizosphere and the number of common bacterial OTUs between the rhizosphere and non-rhizosphere were much lower than in the other seasons, while the number of specific bacterial OTUs in the rhizosphere was much higher than in the other seasons. In the rhizosphere, the number of specific bacterial OTUs in spring and winter was higher than in summer and autumn, while the opposite was observed in the non-rhizosphere. The number of common bacterial OTUs among each season in the rhizosphere was 934, which was higher than the value of 456 in the non-rhizosphere.

As observed with the bacterial OTUs, the number of specific fungal OTUs in the rhizosphere was higher than in the non-rhizosphere, except for in summer. The highest number of common fungal OTUs between the rhizosphere and non-rhizosphere was in summer. In both the rhizosphere and non-rhizosphere, the highest number of specific fungal OTUs was detected in summer, while the lowest was found in autumn. The number of common fungal OTUs among each season in the rhizosphere was 94, which was higher than the value of 43 in the non-rhizosphere.

#### 3.1.3. Microbial Diversity

According to Table 1, there were no significant differences in bacterial Sobs and Shannon indexes among different seasons in the rhizosphere. The bacterial Sobs, Chao 1, and Shannon indexes in winter were much lower than in the other seasons in the non-rhizosphere. The highest fungal Sobs, Chao 1, and Shannon indexes were detected in summer in both the rhizosphere and non-rhizosphere. There was no significant difference in fungal Sobs and Shannon index between the rhizosphere and non-rhizosphere during the entire year. In autumn, the fungal Chao 1 index in the rhizosphere was remarkably higher than in the non-rhizosphere.

PCoA and Permanova (Figure 3) indicated that seasonality had a great influence on the soil bacterial communities in the non-rhizosphere (*p* < 0.001), fungal communities in the rhizosphere (*p* < 0.001), and fungal communities in the non-rhizosphere (*p* < 0.001). Across the whole year, the bacterial communities in the rhizosphere differed significantly (*p* = 0.048) from those in the non-rhizosphere (Figure 3A). However, there was no significant difference (*p* = 0.073) between fungal communities in the rhizosphere and non-rhizosphere (Figure 3B).

#### 3.1.4. Microbial Biomarkers in the Rhizosphere

In order to identify biomarkers in the rhizosphere, LEfSe and indicator analysis were applied. As observed with microbial diversity and composition, the biomarkers in the rhizosphere were also significantly affected by seasonal variation. According to Appendix A, the most bacterial biomarkers were detected in winter, while there were no biomarkers in summer in the rhizosphere. As observed with the bacterial biomarkers, the most fungal biomarkers were detected in winter, while the least was detected in summer (Appendix A).

Across the entire year, the bacterial and fungal biomarkers in the rhizosphere were both much higher than in the non-rhizosphere. In the rhizosphere, the bacterial biomarkers included *Chthoniobacter*, *Rhodanobacter*, *Reyranella*, *Aquicella*, *Subgroup_10*, *Granulicella*, *Labrys*, *Gemmata*, *GAS113*, *Sphingomonas*, and *Pajaroellobacter* at the genus level (Figure 4A), while the fungal biomarkers were *Trechispora*, *Staphylotrichum*, *Chaetosphaeria*, *Trichoderma*, *Mortierella*, *Trichoglossum*, *Lecanicillium*, *Sarcodon*, and *Leptodontidium* at the genus level (Figure 4B).

### 3.2. Functional Analysis of Microorganisms

The abundances of nitrogen metabolism, oxidative phosphorylation, glycine, serine and threonine metabolism, glutathione metabolism, and sulfur metabolism in the rhizosphere were significantly higher than in the non-rhizosphere (Figure 5). According to the FUNGuild analysis, endophytes and ectomycorrhizal fungi in the rhizosphere were significantly higher than in the non-rhizosphere (Figure 6B). Trophic analysis indicated that symbiotrophs were more abundant in the rhizosphere, while pathotrophs were more abundant in the non-rhizosphere (Figure 6B).

### 3.3. Interactions Between Environmental Factors and Microbial Communities

According to Appendix A, seasonality and rhizosphere had a remarkable influence on soil physiochemical properties. Different soil physiochemical properties had different responses to seasonal variation. Available nutrients, such as AP, AK, AMN, NN, and AHN, varied more frequently than TP, TN, and TOC along with seasonal dynamics. AP was higher in spring and summer than in winter and autumn, while TK, AMN, NN, and SWC were higher in autumn and winter than in spring and summer. There was no significant difference in most soil physiochemical properties between the rhizosphere and non-rhizosphere, except for AMN, AHN, TOC, and N/P. AMN was significantly higher in the non-rhizosphere than in the rhizosphere in summer. AHN was significantly lower in the rhizosphere than in the non-rhizosphere in spring, while the result was reversed in winter. TOC and N/P were both significantly higher in the rhizosphere than in the non-rhizosphere in winter.

Appendix A indicates the environmental factors with significant contributions to the microbial communities in the rhizosphere and non-rhizosphere soil at the genus level. According to Table 2, there were no significant correlations between environmental factors and bacterial diversity in the rhizosphere. According to Table 3, in the rhizosphere, fungal abundance was positively correlated with TP and Tem, but negatively correlated with TK, NN, and N/P.

In order to elucidate the interactions between environmental factors and microbial species, the biomarkers and the top 10 abundant species at the genus level were selected to conduct correlation analysis with environmental factors (Appendix A). In both bacteria and fungi, the interactions between microorganisms and environmental factors were strikingly different between the rhizosphere and non-rhizosphere. pH was significantly correlated with most bacteria in the rhizosphere (Appendix A). *Acidothermus,* the most abundant bacteria in the rhizosphere, was significantly positively related to Tem both in the rhizosphere and non-rhizosphere. Compared with bacteria, there were more significant correlations between environmental factors and fungi at the genus level (Appendix A). Most fungi were negatively correlated with Tem, but positively correlated with NN and AHN in the rhizosphere.

## 4. Discussion

Numerous studies have shown that seasonal variation strongly regulates soil physiochemical properties and microorganisms [10,59,60,61,62]. However, no studies have assessed the seasonal variation in soil properties and microbial communities in the rhizosphere of *C. yuhsienensis*. In this study, seasonal variation was the predominant influencing factor on soil physiochemical properties. The soil properties among seasons in this study differed greatly from those of Zhang, Cui, Guo, and Xi [30] in *C. oleifera* forests. This might be because the climate and soil types are markedly different. Many studies have reported that the soil properties in the rhizosphere differ greatly from those in the non-rhizosphere [30,63]. Our result showed that AHN, TOC, and N/P were both significantly higher in the rhizosphere than in the non-rhizosphere in winter. This indicates that the *C. yuhsienensis* roots maintained higher activity in winter. No significant differences were found in soil pH between rhizosphere and non-rhizosphere—which is not consistent with Zhang et al. [30], who found that the rhizosphere pH of *C. oleifera* was significantly lower than non-rhizosphere. A previous study showed that in order to acquire more phosphorous, *C. oleifera* exuded various organic acid, which led to the decline of pH [64]. These results suggested that *C. yuhsienensis* roots might not be as active as *C. oleifera* and had a weaker influence on soil physiochemical properties. 

Compared with bacteria, fungi were more sensitive to soil physicochemical properties in this study (Table 2 and Table 3). The positive correlations between TP and fungal richness (Table 3) in this study are contrary to those studies which showed that phosphorous is the key limiting factor on the establishment of symbioses between plant and mycorrhizal fungi [65,66]. Previous studies about the negative influences of phosphorous on mycorrhizal fungi were under the high content of phosphorous. However, the contents of the AP and TP were low in this study (Appendix A), which is not enough to inhibit mycorrhizal fungi. AP was widely considered to be a limiting factor of yield for oil tea in China [67,68]. Otherwise, mycorrhizal fungi were not dominant in the fungal community in this study, which also explained why phosphorous contents were positively correlated with fungal richness. 

The dominant bacteria, Chloroflexi, Proteobacteria, Acidobacteria, Actinobacteria, and Planctomycetes, and fungi, Ascomycota and Basidiomycota, in the rhizosphere of *C. yuhsienensis* were similar to previous studies in *C. oleifera* rhizosphere or bulk soil [10,30], but a little different from the rhizosphere soil of *Camellia sinensis* [69]. These results indicated that nutrient utilization patterns of the plant might significantly alter microbial communities. There were significantly different responses between the rhizosphere and non-rhizosphere microbial communities to seasonal variation. During the entire year, the shared OTUs of the four seasons in the rhizosphere were higher than those in the non-rhizosphere for both bacteria and fungi (Figure 2). The PCoA also indicated that the effect of seasonal variation on bacterial community structure was smaller in the rhizosphere (R^2^ = 0.328, *p* = 0.107) than in the non-rhizosphere (R^2^ = 0.398, *p* < 0.001) (Figure 3A). On account of the root activity of *C. yuhsienensis*, the bacterial community structure in the rhizosphere differed significantly from that in the non-rhizosphere (R^2^ = 0.064, *p* = 0.048). Particularly in winter, the bacterial OTUs in the rhizosphere did not decrease due to the low temperature, as observed in the non-rhizosphere, but rather the number of exclusive OTUs increased (Figure 2A). The bacterial Sobs, Chao 1, and Shannon indexes were significantly higher in the rhizosphere than in the non-rhizosphere in winter. These results indicated that the roots of *C. yuhsienensis* might have higher excretions in winter, thus remarkably increasing the activity of bacteria in winter. This result is consistent with past studies that indicated that the roots of *C. yuhsienensis* are also active in winter [70,71]. The fungal community structures also varied greatly with seasonal variation (Figure 3B). Across the entire year, fungi in the non-rhizosphere were more sensitive to seasonal variation than those in the rhizosphere (Table 1 and Figure 2B). Notably, there were slight differences between the rhizosphere and non-rhizosphere microbial communities, regardless of microbial community structure, microbial diversity, or microbial OTUs (Figure 1 and Figure 2 and Table 1), and the number of fungal and bacterial biomarkers was lowest in summer in both the rhizosphere and non-rhizosphere (Appendix A and Appendix A). In summary, seasonal variation was the predominant effective factor influencing microbial diversity and structure, and the activity of *C. yuhsienensis* roots could provide a more stable environment for microorganisms, especially for bacteria in winter. 

Exploring the rhizosphere microbial function is a great way to understand the interactions between plants and microorganisms. Although there are already some researchers focusing on microbial communities associated with oil tea [10,30], the microbial functions in the oil tea rhizosphere have never been noticed. Indicator analysis showed that the number of biomarkers in the rhizosphere was much higher than in the non-rhizosphere (Figure 4). All of the bacterial biomarkers in the rhizosphere in this study were chemoorganotrophic Gram-negative bacteria that grow under aerobic conditions [72,73,74,75,76,77,78]. Some species of *Rhodanobacter* can degrade γ-hexachlorocyclohexane (lindane; γ-HCH) under aerobic conditions, and some species have the ability to antagonize root rot fungi [79] or to denitrify nitrate [80,81]. Most *Labrys* species have been reported as biodegraders of some harmful organic matter for animals, such as *Labrys portucalensis* [82,83]. *Sphingomonas*, a Gram-negative bacteria that was a biomarker in the rhizosphere, grows under aerobic conditions. *Sphingomonas* species are widely distributed throughout the soil and water and are associated with plant roots [84]. *Sphingomonas* species are multifunctional. Some exhibit antagonism against the phytopathogenic fungus *Verticillium dahliae* [85], while some are the pathogenic bacteria causing plant root disease [86], and some are able to degrade refractory pollutants [87]. The biomarkers reflect that the rhizosphere environment of *C. yuhsienensis* had high air permeability and abundant organic nutrients for these bacteria.

Most fungal biomarkers in the rhizosphere of *C. yuhsienensis* are beneficial fungi for plants. *Trichoderma* has been reported several times to protect plants and contain pathogen populations [88,89]. This is the first study to report *Trichoderma* in the rhizosphere of *C. yuhsienensis*. *Mortierella* is a large and diverse genus, and most species of *Mortierella* are saprobic and abundant in the soil and plant debris [90]. Arachidonic acid (ARA), produced by most *Mortierella* species, is an important constituent of biological cells [91]. ARA is not only beneficial for animals [92], but can also antagonize plant disease [93]. *Lecanicillium* species have great potential as biocontrol agents [94]. Most *Lecanicillium* species are able to parasitize many insects, such as *Lecanicillium longisporum* against aphids, *Lecanicillium muscarium* against whiteflies and thrips [95], and *Lecanicillium psalliotae* against the root-knot nematode *Meloidogyne incognita* [96]. *Meloidogyne* incognita is one of the major nematodes of oil tea roots [97].

In general, most biomarkers in the rhizosphere of *C. yuhsienensis* are beneficial for plants, and thus, we may speculate that the root activity of *C. yuhsienensis* markedly improved the rhizosphere conditions and increased the beneficial microorganisms. Functional analysis also verified this (Figure 5 and Figure 6).

## 5. Conclusions

The dominant bacteria in the rhizosphere of *C. yuhsienensis* were Chloroflexi, Proteobacteria, Acidobacteria, Actinobacteria, and Planctomycetes, and the dominant fungi were Ascomycota, Basidiomycota, and Mucoromycota. Seasonal variation was the predominant factor influencing soil microbial abundance, diversity, and structure and the soil physiochemical properties in the rhizosphere of *C. yuhsienensis*. There were weak correlations between microorganisms and soil physiochemical properties in the rhizosphere. Seasonal variation had significant effects on bacterial abundance, but not on diversity and structure, and had significant effects on fungal abundance and structure, but not on diversity in the rhizosphere. Significantly higher bacterial abundance and diversity were observed in the rhizosphere than in the non-rhizosphere, which reflected the high activity of the *C. yuhsienensis* roots in winter. All of the bacterial biomarkers in the rhizosphere were chemoorganotrophic Gram-negative bacteria that grow under aerobic conditions, and the fungal biomarkers, including *Trichoderma*, *Mortierella,* and *Lecanicillium*, have a great ability to protect against pathogens in the rhizosphere. In the rhizosphere of *C. yuhsienensis*, the dominant functions of the bacteria were oxidative phosphorylation and metabolism of nitrogen, glycine, serine, threonine, glutathione, and sulfur in the rhizosphere, while the dominant fungal functional groups were endophytes and ectomycorrhizal fungi of the symbiotroph trophic type. The activity of *C. yuhsienensis* created better conditions with good air permeability and formed a microbial community dominated by beneficial bacteria and fungi. We strongly suggest that future research should pay greater attention to the root vitality and exudation of *C. yuhsienensis* in different seasons, which might be the key to a better understanding of the relationships between roots and microbial community. This is the first time to study *C. yuhsienensis* rhizosphere microbial communities during seasonal variations, and the first time to study *Camellia* species rhizosphere microbial function. Therefore, we strongly suggest that researchers pay more attention to the rhizosphere microorganisms and provide more scientific data to better study and manage *Camellia* plantations. 

## Figures and Tables

**Figure 1 microorganisms-08-01385-f001:**
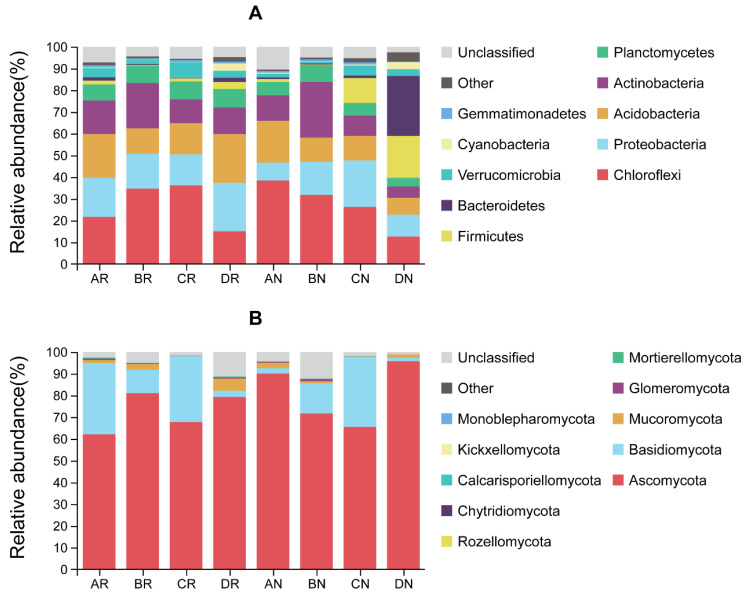
Relative abundance of bacteria (**A**) and fungi (**B**) at the phylum level. AR, BR, CR, and DR indicates rhizosphere soil sample in spring, summer, autumn, and winter, respectively. AN, BN, CN, and DN indicates non-rhizosphere soil sample in spring, summer, autumn, and winter, respectively. *n* = 3.

**Figure 2 microorganisms-08-01385-f002:**
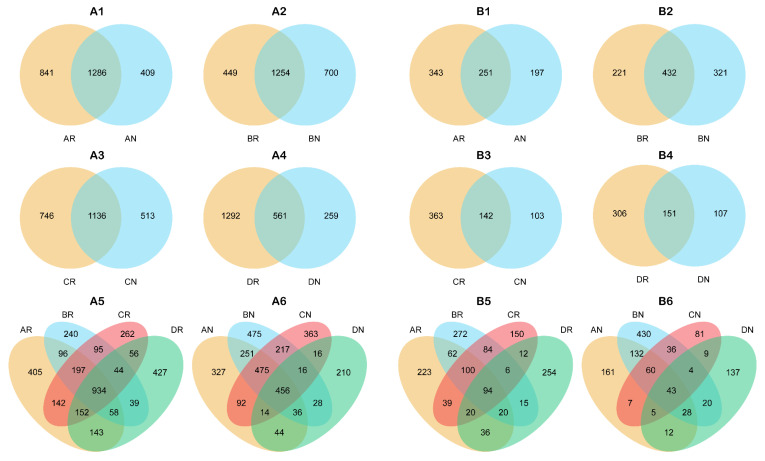
Venn diagram of bacterial (**A1**–**A6**) and fungal (**B1**–**B6**) Operational taxonomic units (OTUs). AR, BR, CR, and DR indicates rhizosphere soil sample in spring, summer, autumn, and winter, respectively. AN, BN, CN, and DN indicates non-rhizosphere soil sample in spring, summer, autumn, and winter, respectively. Overlaps and non-overlaps indicate shared and exclusive OTUs under different samples, respectively. *n* = 3.

**Figure 3 microorganisms-08-01385-f003:**
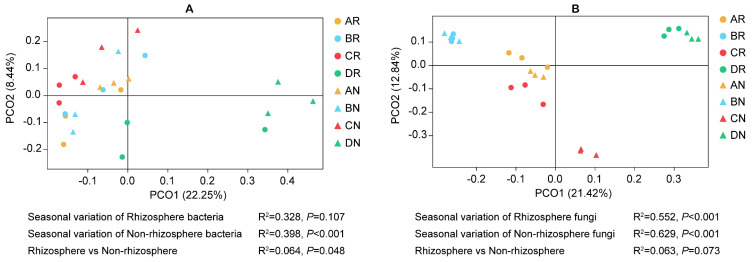
Principal Co-ordinary Analysis (PCoA) of bacterial (**A**) and fungal (**B**) community based on unweighted unifrac distance. AR, BR, CR, and DR indicates rhizosphere soil sample in spring, summer, autumn, and winter, respectively. AN, BN, CN, and DN indicates non-rhizosphere soil sample in spring, summer, autumn, and winter, respectively. Values of R2 and P were calculated using Permutational MANOVA (Permanova). *n* = 3.

**Figure 4 microorganisms-08-01385-f004:**
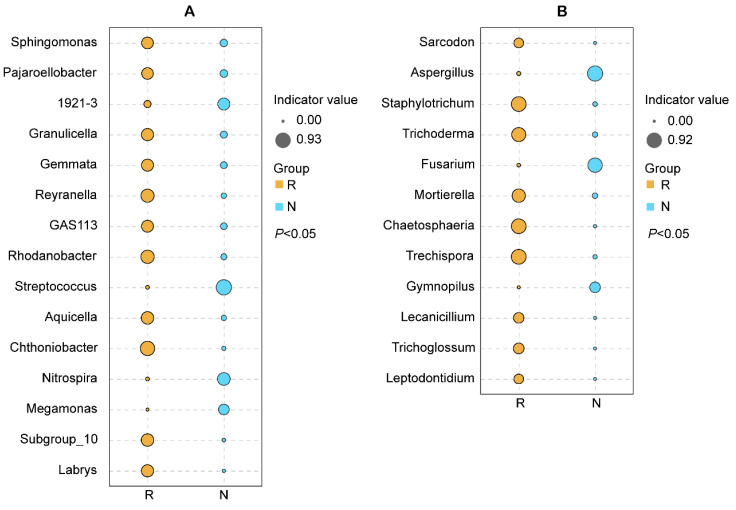
Bacterial (**A**) and fungal (**B**) biomarkers of the rhizosphere (R) and non-rhizosphere (N) samples at the genus level based on indicator analysis. The size of the cycle indicates the indicator value of each genus. *n* = 3.

**Figure 5 microorganisms-08-01385-f005:**
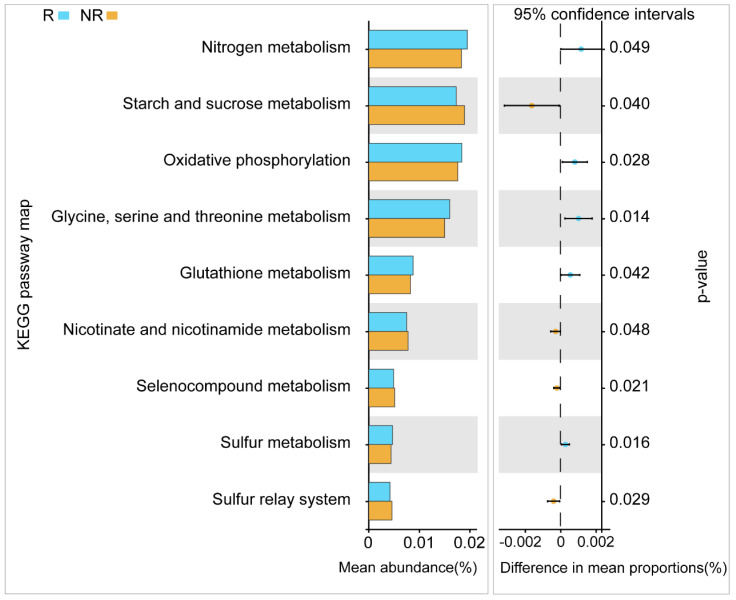
The difference Kyoto Encyclopedia of Genes and Genomes (KEGG) map of bacteria between rhizosphere (R) and non-rhizosphere (NR) based on Welch’s t-test. *n* = 3.

**Figure 6 microorganisms-08-01385-f006:**
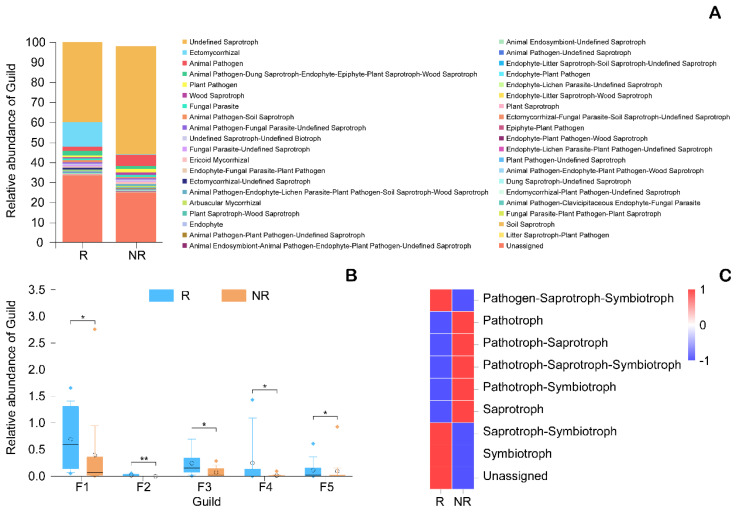
Distributed stack diagram (**A**) of fungal functional group (Guild), box figure of different fungal functional group (Guild) based on Wilcoxon rank-sum test (**B**), and the heatmap of fungal functional trophic mode (**C**). R and NR indicated the rhizosphere and non-rhizosphere, respectively. F1, F2, F3, F4, and F5 indicated Fungal Parasite-Undefined Saprotroph, Ectomycorrhizal-Undefined Saprotroph, Endophyte-Litter Saprotroph-Soil Saprotroph-Undefined Saprotroph, Endophyte-Litter Saprotroph-Wood Saprotroph, and Ectomycorrhizal-Fungal Parasite-Soil Saprotroph-Undefined Saprotroph. * and ** over the box indicated a significant difference at 0.05 and 0.01 level, respectively. The black line and hollow cycle in the box indicated the median and mean (*n* = 3) abundance of Guild, respectively. The solid rhombus beyond the box indicated the discrete data. The top and down line of box indicated 25% and 75% of Guild. The top and down line beyond the box indicated 10% and 90% of Guild.

**Table 1 microorganisms-08-01385-t001:** Alpha diversity (Sobs, Shannon, and Chao 1 index) of bacteria and fungi.

Kingdom	Sample	Sobs	Chao 1	Shannon
Bacteria	AR	1780 ± 390 a	2218.8 ± 434.27 a	8.12 ± 0.1 a
	BR	1392 ± 395 a	1757.83 ± 330.26 abc	7.31 ± 0.74 ab
	CR	1727 ± 116 a	2165.43 ± 222.65 ab	7.72 ± 0.44 ab
	DR	1271 ± 398 a	1540.55 ± 410.06 c	7.97 ± 0.24 a
	AN	1422 ± 159 a	1857.35 ± 83.64 abc	7.69 ± 0.42 ab
	BN	1641 ± 314 a	1973.68 ± 340.82 abc	7.62 ± 0.82 ab
	CN	1265 ± 324 a	1672.15 ± 199.73 bc	7.6 ± 1.05 ab
	DN	609 ± 99 b	882.488 ± 105.93 d	6.64 ± 0.74 b
Fungus	AR	497 ± 113 ab	683.65 ± 67.99 ab	4.34 ± 1.41 bc
	BR	609 ± 109 a	803.52 ± 94.73 a	5.81 ± 0.45 ab
	CR	423 ± 84 abc	597.05 ± 54.6 abc	3.24 ± 2.31 c
	DR	417 ± 25 abc	584.27 ± 22.39 abcd	5.29 ± 0.17 abc
	AN	389 ± 80 abc	538.32 ± 114.33 bcd	5.08 ± 0.1 bc
	BN	620 ± 133 a	759.81 ± 154.94 ab	6.36 ± 0.05 a
	CN	213 ± 32 c	315.57 ± 26.62 d	3.85 ± 1.56 bc
	DN	299 ± 21 bc	451.79 ± 6.08 cd	3.16 ± 0.27 c

AR, BR, CR, and DR indicates rhizosphere soil sample in spring, summer, autumn, and winter, respectively. AN, BN, CN, and DN indicates non-rhizosphere soil sample in spring, summer, autumn, and winter, respectively. Numbers before and behind “±” are mean value (*n* = 3) and SE, respectively. Lowercases behind numbers indicate a significant difference.

**Table 2 microorganisms-08-01385-t002:** Correlations between bacterial alpha-diversity and environmental factors based on Pearson correlation coefficients with the two-tail test.

Environmental Factors	Rhizosphere			Non-Rhizosphere		
	Sobs	Chao 1	Shannon	Sobs	Chao 1	Shannon
AP	0.068	0.098	−0.150	0.388	0.452	0.260
TP	−0.130	−0.197	0.119	0.437	0.351	0.211
AK	0.069	0.105	0.512	0.532	0.521	0.281
TK	0.252	0.140	0.303	−0.198	−0.187	0.017
AMN	0.273	0.252	0.276	−0.436	−0.492	−0.180
NN	−0.161	−0.159	−0.099	−0.788 **	−0.822 **	−0.379
AHN	−0.471	−0.485	−0.296	−0.229	−0.169	−0.258
TN	−0.117	−0.018	−0.171	0.540	0.617 *	0.357
TOC	−0.097	−0.041	0.220	0.163	0.168	−0.054
pH	0.267	0.315	0.118	0.270	0.282	0.471
SWC	0.360	0.457	0.356	−0.556	−0.491	−0.358
Tem	−0.121	−0.166	0.137	0.849 **	0.866 **	0.491
C/N	−0.110	−0.126	0.272	−0.336	−0.402	−0.372
C/P	−0.096	0.017	−0.015	−0.148	−0.073	−0.237
N/P	−0.089	−0.013	−0.230	0.347	0.454	0.282

“*” and “**” indicate the significant correlations at 0.05 and 0.01 level, respectively. AP, available phosphorous; TP, total phosphorous; AK, available potassium; TK, total potassium; AMN, ammonium nitrogen; NN, nitrate nitrogen; AHN, alkaline hydrolyzable nitrogen; TN, total nitrogen; TOC, total organic carbon; SWC, soil water content; Tem, soil monthly mean temperature; C/N, ratio of total carbon to total nitrogen; C/P, ratio of total carbon to total phosphorous; N/P, ration of total nitrogen to total phosphorous. *n* = 12.

**Table 3 microorganisms-08-01385-t003:** Correlations between environmental factors and fungal alpha-diversity based on Pearson correlation coefficients with the two-tail test at 0.05 (*) and 0.01 (**) level.

Environmental Factors	Rhizosphere			Not-Rhizosphere		
	Sobs	Chao 1	Shannon	Sobs	Chao 1	Shannon
AP	0.210	0.273	−0.006	0.255	0.277	0.414
TP	0.760 **	0.758 **	0.145	0.660 *	0.586 *	0.582 *
AK	0.183	0.239	0.057	0.475	0.471	0.464
TK	−0.551	−0.703 *	−0.181	−0.514	−0.563	−0.510
AMN	−0.432	−0.555	−0.391	−0.507	−0.561	−0.543
NN	−0.521	−0.620 *	0.099	−0.479	−0.387	−0.696 *
AHN	−0.411	−0.440	0.219	−0.411	−0.412	−0.259
TN	−0.229	−0.122	−0.041	0.197	0.266	0.289
TOC	−0.339	−0.253	0.163	0.389	0.434	0.226
pH	0.311	0.348	0.000	−0.071	−0.091	0.003
SWC	−0.419	−0.432	−0.242	−0.683 *	−0.603 *	−0.776 **
Tem	0.550	0.654 *	−0.012	0.541	0.448	0.738 **
C/N	−0.296	−0.234	0.187	0.205	0.200	−0.056
C/P	−0.550	−0.492	0.010	−0.025	0.099	−0.198
N/P	−0.637 *	−0.601 *	−0.123	−0.108	−0.027	0.023

“*” and “**” indicate the significant correlations at 0.05 and 0.01 level, respectively. AP, available phosphorous; TP, total phosphorous; AK, available potassium; TK, total potassium; AMN, ammonium nitrogen; NN, nitrate nitrogen; AHN, alkaline hydrolyzable nitrogen; TN, total nitrogen; TOC, total organic carbon; SWC, soil water content; Tem, soil monthly mean temperature; C/N, ratio of total carbon to total nitrogen; C/P, ratio of total carbon to total phosphorous; N/P, ration of total nitrogen to total phosphorous. *n* = 12.

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
