# Peer review of "Seasonal Variation in the Rhizosphere and Non-Rhizosphere Microbial Community Structures and Functions of *Camellia yuhsienensis* Hu"

_microorganisms, 2020, doi:10.3390/microorganisms8091385_

Round 1
Reviewer 1 Report
The authors provide an interesting and well-written insight into the seasonal variations impacting the microorganisms in the rhizosphere of Camellia yuhsienensis. Understanding this is crucial to the growth and pathogen resistance of any crop. Below are my comments:
Line 126 – Were negative controls used in the study? If so, please include some information about how they performed and if their results had any implications on the dataset used in the study.
Line 144 – Please include the BioProject and other relevant accessions from SRA in the paper.
Line 151 – Please explicitly state what kind of clustering was done. If it was reference based, what database was used?
Author Response
Response to Reviewer 1 Comments
Dear Editor and Reviewer:
Thank you for your letter and comments concerning our manuscript entitled “Seasonal variation in the microbial community structures and functions of Camellia yuhsienensis Hu” (ID: microorganisms-912123). Those comments are all valuable and very helpful for revising and improving our paper, and important to guide our future research. We have studied comments carefully and have made correction which we hope to satisfy reviewers and editors. We used the track changes mode in MS word. The main corrections and the responses to the editor’s and reviewer’s comments are as flowing:
Point 1: Line 126 – Were negative controls used in the study? If so, please include some information about how they performed and if their results had any implications on the dataset used in the study.
Response 1: The only control is non-rhizosphere soil in this study. As this is an investigative article, the negative controls were not used in this study. After reviewing references, we found that there is usually no negative control in the researches which is similar to our study. For example, Zhang et al., (2020), Characteristics of the soil microbial community in the forestland of Camellia oleifera. We hope this explanation can answer your question.
Point 2: Line 144 – Please include the BioProject and other relevant accessions from SRA in the paper.
Response 2: Thanks for your reminding. Line147 – The BioProject (PRJNA646017) is included in the manuscript as reviewer advised.
Point 3: Line 151 – Please explicitly state what kind of clustering was done. If it was reference based, what database was used?
Response 3: Line 153-154 – The clusters are OTUs as mentioned the last sentence “The effective tags were clustered into operational taxonomic units (OTUs) of ≥ 97% similarity using UPARSE (version 9.2.64) pipeline”. The reference is “Edgar, (2013), UPARSE: Highly accurate OTU sequences from microbial amplicon reads” which had been cited in the manuscript.

Reviewer 2 Report
Presented manuscript reveals the specificity of microbial communities of rhizosphere and non-rhizosphere soil microbial communities associated with Camellia yuhsienensis.
Major comment:
I think, that Discussion section is incompletely: several paragraphs on reasons of revealed functional activity of rhizosphere communities and reasons of its' seasonal dynamics would be useful. How does the observed results on microbial communities could be explained from the points of Camellia yuhsienensis physiology and biochemistry? Also, it seems to be interesting to compare the revealed both taxonomical and functional diversity of the studied communities with some data on the rhizospheric microbial communities of the other Camellia species, if it is possible.
Minor comments:
Line 44 - "Venturia inaequalis" should be italic.
Line 77 - I'd recommend to clarify the resistance of Camellia sinensis in this sentence: resistance to drought and seasonal environmental changes, resistance to plant diseases etc. For instance, it could be done by the colon after reference 29.
Line 144 - accession number in NCBI SRA database should be added.
Line 179 - URL should formatted as reference.
Line 354 - I'd recommend to correct "Indicator analysis showed..." by literary reasons
Line 357 - Rhodanobacter (genus) should be italic.
References should be revised for doi formatting: somewhere (Ref 8, 27, 43) doi is mentioned twice; in many references "doi:" before the link is missed.
Author Response
Response to Reviewer 2 Comments
Dear Editor and Reviewer:
Thank you for your letter and comments concerning our manuscript entitled “Seasonal variation in the microbial community structures and functions of Camellia yuhsienensis Hu” (ID: microorganisms-912123). Those comments are all valuable and very helpful for revising and improving our paper, and important to guide our future research. We have studied comments carefully and have made correction which we hope to satisfy reviewers and editors. We used the track changes mode in MS word. The main corrections and the responses to the editor’s and reviewer’s comments are as flowing:
Point 1: I think, that Discussion section is incompletely: several paragraphs on reasons of revealed functional activity of rhizosphere communities and reasons of its' seasonal dynamics would be useful. How does the observed results on microbial communities could be explained from the points of Camellia yuhsienensis physiology and biochemistry? Also, it seems to be interesting to compare the revealed both taxonomical and functional diversity of the studied communities with some data on the rhizospheric microbial communities of the other Camellia species, if it is possible.
Response 1: Thanks for your advices. Line 328-347 – We added some content about taxonomical diversity comparisons Camellia yuhsienensis with other Camellia species. However, only few studies focused on rhizosphere microorganisms and no one has studied the microbial function in Camellia species rhizosphere until now. So we could not compare the microbial function with others.
Point 2: Line 44 - "Venturia inaequalis" should be italic.
Response 2: Thanks for your reminding. Line 46 – The format of "Venturia inaequalis" has been revised as italic in the manuscript.
Point 3: Line 77 - I'd recommend to clarify the resistance of Camellia yuhsienensis in this sentence: resistance to drought and seasonal environmental changes, resistance to plant diseases etc. For instance, it could be done by the colon after reference 29.
Response 3: Thanks for your advices. Line 80 – The sentence was changed to “Camellia yuhsienensis Hu, a species of oil tea, was once widely cultivated in central China because of its high yield and high resistance to disease pathogens” as reviewer advised.
Point 4: Line 144 - accession number in NCBI SRA database should be added.
Response 4: Thanks for your reminding. Line 147 – The BioProject (PRJNA646017) is included in the manuscript as reviewer advised.
Point 5: Line 179 - URL should formatted as reference.
Response 5: Thanks for your advice. Line 182 – The URL has been formatted as reference in the manuscript.
Point 6: Line 354 - I'd recommend to correct "Indicator analysis showed..." by literary reasons
Response 6: Thanks for your advice. Line 373 – “Indicator analysis indicated” has been revised as “Indicator analysis showed”
Point 7: Line 357 - Rhodanobacter (genus) should be italic.
Response 7: Thanks for your reminding. Line 376 – The format of " Rhodanobacter" has been revised as italic in the manuscript.
Point 8: References should be revised for doi formatting: somewhere (Ref 8, 27, 43) doi is mentioned twice; in many references "doi:" before the link is missed.
Response 8: Thanks for your reminding. The repetitive doi have been deleted and the missing “doi” have been added.
Reviewer 3 Report
The manuscript describes an analysis of the rhizospheric and non-rhizospheric microbial community of Camellia yuhsienensis. The work is interesting, well organized and well written.
I have some suggestions and comments:
Lines 129-130. You used two very short primers for DNA amplification. It is preferable to use primers with sequences of at least 50 bases. For example, we are using 16S Amplicon PCR Forward Primer = 5 'TCGTCGGCAGCGTCAGATGTGTATAAGAGACA GCCTACGGGNGGCWGCAG, 16S Amplicon PCR Reverse Primer = 5' GTCTCGTGGGCTCG GAGATGTGTATAAGAGACAGGATACHVGGGT. They work very well and allow you to isolate even the archaea.
The only part that does not convince me very much concerns the influence of some environmental factors on the microbial community. For example, it is strange that fungi are negatively correlated with nitrogen and positively correlated with P. P can inhibit the establishment of mycorrhizal symbiosis.
In the discussion section you comment on the tables in the supplementary materials and do not comment, or very shortly, on tables 2 and 3. Please add some comment.
Lines 13 and 32. “Microbial flora”. It is more correct to use "microbial community". Microbial flora is an old and incorrect term.
Table 1. Change "fungus" in the plural "fungi".
Author Response
Response to Reviewer 3 Comments
Dear Editor and Reviewer:
Thank you for your letter and comments concerning our manuscript entitled “Seasonal variation in the microbial community structures and functions of Camellia yuhsienensis Hu” (ID: microorganisms-912123). Those comments are all valuable and very helpful for revising and improving our paper, and important to guide our future research. We have studied comments carefully and have made correction which we hope to satisfy reviewers and editors. We used the track changes mode in MS word. The main corrections and the responses to the editor’s and reviewer’s comments are as flowing:
Point 1: Lines 129-130. You used two very short primers for DNA amplification. It is preferable to use primers with sequences of at least 50 bases. For example, we are using 16S Amplicon PCR Forward Primer = 5 'TCGTCGGCAGCGTCAGATGTGTATAAGAGACA GCCTACGGGNGGCWGCAG, 16S Amplicon PCR Reverse Primer = 5' GTCTCGTGGGCTCG GAGATGTGTATAAGAGACAGGATACHVGGGT. They work very well and allow you to isolate even the archaea.
Response 1: Thanks for your advices. Line 132-133 – The primers in this study are the most common used in recent years. Previous researches which used these primers got good results. In our study, the dilution curves showed that the sequencing result meets our research requirements. We will try the primers which are provided by reviewer to conduct the next researches in the future.
Point 2: The only part that does not convince me very much concerns the influence of some environmental factors on the microbial community. For example, it is strange that fungi are negatively correlated with nitrogen and positively correlated with P. P can inhibit the establishment of mycorrhizal symbiosis.
Response 2: As reviewer mentioned, P can inhibit the establishment of mycorrhizal symbiosis. However, after reading references we found that this kind of inhibition is under high P content condition. For example, the reference “Abbott et al., (1984), The effect of phosphorus on the formation of hyphae in soil by the vesicular arbuscular mycorrhizal fungus, Glomus fasciculatum” showed that the length of hyphae was not limited until adding 500 mg/g superphosphate, equivalent of 132 mg/kg P, which is much higher than the available phosphorous content in our study. The area of Camellia oil plantation in Central South of China is lack of phosphorous, including our study site. Otherwise, the fungi in this study are not predominant by mycorrhizal fungi. Therefore, we think the positive correlations between P and fungi are reasonable. We hope these explanations will answer the doubts of the reviewer.
Point 3: In the discussion section you comment on the tables in the supplementary materials and do not comment, or very shortly, on tables 2 and 3. Please add some comment.
Response 3: Thanks for your advices. Line 328-347 – The comments about table 2 and 3 were added in the manuscript as reviewer advised.
Point 4: Lines 13 and 32. “Microbial flora”. It is more correct to use "microbial community". Microbial flora is an old and incorrect term.
Response 4: Thanks for your advices. Lines 14, 35, 89, 419 and 422 – “Microbial flora” in Abstract, Introduction and Conclusion has been revised as “microbial community”
Point 5: Table 1. Change "fungus" in the plural "fungi".
Response 5: “fungus” in table 1 has been revised as “fungi”

Reviewer 4 Report
The manuscript is based on research on soil microbiomes of Camellia yuhsienensis, a Chinese crop popular for its high yield and pathogen resistance. The authors examined the diversity of bacteria and fungi in soil rhizosphere and non-rhizosphere in this crop and identified differences in its function. In addition, the authors found that they had seasonal variations and described the results in the paper. This paper shows meaningful results to many scientists who refer to this journal, but please read the opinions below and consider revising the paper if possible.
The paper not only looks at seasonal variation in the soil of microbial community structures and functions of Camellia yuhsienensis, but also makes comparisons of microbial flora between rhizosphere and non-rrhizosphere soil samples as one of the important research topics. However, the title only covers seasonal variation and the term rhizosphere does not appear in the title, which can be misleading for readers. Please consider this and modify the title.
While the purpose of the study in Abstract is "to explore the influence of seasonal variation on the microbial community in the rhizosphere of C. yuhsienensis using high-throughput sequencing”(lines 13-15), the first line of the conclusion in Abstract is “there were significant differences in microorganisms and their functions between the rhizosphere and non-rhizosphere of C. yuhsienensis”(lines 28-29). They do not seem to respond to each other. Of course, the authors are talking about seasonal variations after this sentence, but I think they need to make corrections for better understanding of the reader.
The main theme of this study is the seasonal variation of microbiome in the soil of Camellia yuhsienensis and the comparison of microbiome in rhizosphere and non-rrhizosphere. The soil microbiome study of Camellia yuhsienensis has not been carried out so far, but this topic (seasonal variation and comparison of rhizosphere vs non-r.) has been studied very much. Therefore, it is necessary to think more about what originality this study has, and to clearly describe it for the purpose of the study. Similarly, the conclusions also need to tell us what the differences between this study and the previous comparative (seasonal; r. vs non-r.) studies of soil microbiomes are, and what is new from this study.
Author Response
Response to Reviewer 4 Comments
Dear Editor and Reviewer:
Thank you for your letter and comments concerning our manuscript entitled “Seasonal variation in the microbial community structures and functions of Camellia yuhsienensis Hu” (ID: microorganisms-912123). Those comments are all valuable and very helpful for revising and improving our paper, and important to guide our future research. We have studied comments carefully and have made correction which we hope to satisfy reviewers and editors. We used the track changes mode in MS word. The main corrections and the responses to the editor’s and reviewer’s comments are as flowing:
Point 1: The paper not only looks at seasonal variation in the soil of microbial community structures and functions of Camellia yuhsienensis, but also makes comparisons of microbial flora between rhizosphere and non-rhizosphere soil samples as one of the important research topics. However, the title only covers seasonal variation and the term rhizosphere does not appear in the title, which can be misleading for readers. Please consider this and modify the title.
Response 1: Thanks for your advices. The title has been revised as “Seasonal variation in the rhizosphere and non-rhizosphere microbial community structures and functions of Camellia yuhsienensis Hu”
Point 2: While the purpose of the study in Abstract is "to explore the influence of seasonal variation on the microbial community in the rhizosphere of C. yuhsienensis using high-throughput sequencing”(lines 13-15), the first line of the conclusion in Abstract is “there were significant differences in microorganisms and their functions between the rhizosphere and non-rhizosphere of C. yuhsienensis”(lines 28-29). They do not seem to respond to each other. Of course, the authors are talking about seasonal variations after this sentence, but I think they need to make corrections for better understanding of the reader.
Response 2: Thanks for your advices. Line 29-33 – The first sentence of conclusion in Abstract has been revised as “In conclusion, seasonal variation had a remarkable influence on the microbial communities and function which were also significantly different in rhizosphere and non-rhizosphere of C. yuhsienensis.”
Point 3: The main theme of this study is the seasonal variation of microbiome in the soil of Camellia yuhsienensis and the comparison of microbiome in rhizosphere and non-rhizosphere. The soil microbiome study of Camellia yuhsienensis has not been carried out so far, but this topic (seasonal variation and comparison of rhizosphere vs non-r.) has been studied very much. Therefore, it is necessary to think more about what originality this study has, and to clearly describe it for the purpose of the study. Similarly, the conclusions also need to tell us what the differences between this study and the previous comparative (seasonal; r. vs non-r.) studies of soil microbiomes are, and what is new from this study.
Response 3: Thanks for your advices. We revised the Abstract, Discussion and Conclusion to clarify the innovation and purpose of the study.
